# A COVID-19 Infection Model Considering the Factors of Environmental Vectors and Re-Positives and Its Application to Data Fitting in Japan and Italy

**DOI:** 10.3390/v15051201

**Published:** 2023-05-19

**Authors:** Shimeng Dong, Jinlong Lv, Wanbiao Ma, Boralahala Gamage Sampath Aruna Pradeep

**Affiliations:** 1Department of Applied Mathematics, School of Mathematics and Physics, University of Science and Technology Beijing, Beijing 100083, China; 2Department of Mathematics, University of Ruhuna, Matara 81000, Sri Lanka

**Keywords:** COVID-19 infection, asymptomatic infection/re-positive, environmental vector, stability, data fitting

## Abstract

COVID-19, which broke out globally in 2019, is an infectious disease caused by a novel strain of coronavirus, and its spread is highly contagious and concealed. Environmental vectors play an important role in viral infection and transmission, which brings new difficulties and challenges to disease prevention and control. In this paper, a type of differential equation model is constructed according to the spreading functions and characteristics of exposed individuals and environmental vectors during the virus infection process. In the proposed model, five compartments were considered, namely, susceptible individuals, exposed individuals, infected individuals, recovered individuals, and environmental vectors (contaminated with free virus particles). In particular, the re-positive factor was taken into account (i.e., recovered individuals who have lost sufficient immune protection may still return to the exposed class). With the basic reproduction number $R_{0}$ of the model, the global stability of the disease-free equilibrium and uniform persistence of the model were completely analyzed. Furthermore, sufficient conditions for the global stability of the endemic equilibrium of the model were also given. Finally, the effective predictability of the model was tested by fitting COVID-19 data from Japan and Italy.

## 1. Introduction

In 2019, an unexplained novel coronavirus disease (COVID-19) suddenly broke out, which triggered an unprecedented public health crisis in the world. On 11 March 2020, the World Health Organization (WHO) declared COVID-19 as a global pandemic. COVID-19 is caused by a novel coronavirus named severe acute respiratory syndrome coronavirus 2 (SARS-CoV-2) [1]. SARS-CoV-2 is regarded as the third zoonotic coronavirus emerging in the current century, after SARS-CoV in 2002 and the Middle East respiratory syndrome coronavirus (MERS-CoV) in 2012 [2,3]. Common signs of COVID-19 infection include respiratory symptoms, fever, cough, shortness of breath, and difficulty of breathing, and in more serious cases, it can lead to pneumonia, severe acute respiratory syndrome, renal failure, and even death due to alveolar damage, posing a serious threat to human life [4,5]. Among the low-risk people who died of COVID-19, 70% of them had one or more organ dysfunctions within 4 months of the initial symptoms of COVID-19, and experienced organ damage, fatigue, muscle weakness, difficulty of sleeping, anxiety, or depression after the acute phase [6,7]. On a global scale, as of 24 May 2022, countries around the world have reported more than 500 million confirmed cases of COVID-19 to the WHO, including 6.2 million deaths cases [8].

It is encouraging that in the past two years, with the joint efforts of scientists from many countries, a series of important research achievements have been made in the field of drugs for COVID-19 treatment, improving the effectiveness of vaccines, which have greatly contributed to protecting people’s health and lives [9,10].

In particular, differential equations and statistics have been widely used to construct various types of compartmental models and make rapid and effective predictions about infection trends, epidemic patterns, and key factors in the spread of infectious diseases. In addition, mathematical models based on differential equations provide a theoretical basis and optimal strategies for the prevention and treatment of infectious diseases [11,12,13,14,15,16,17,18,19,20,21,22,23,24,25,26,27].

It is known that environmental vectors play an important role in virus infection and transmission, and bring new difficulties and challenges to the prevention and control of the disease. For example, a study reviewing 22 types of coronaviruses revealed that viruses such as SARS-CoV, SARS-CoV, MERS-CoV, and endemic human coronaviruses can persist for up to 9 days on inanimate surfaces such as metal, glass, or plastic [28]. Another experimental study found that SARS-CoV-2 can be detected on aerosols for up to 3 h, on copper for up to 4 h, on cardboard for up to 24 h, and on plastic and stainless steel for up to 3 days [29]. Moreover, SARS-CoV-2 can also be detected in domestic wastewater [30,31]. In recent years, many scholars have conducted extensive research on infectious disease models with media effects [32,33,34,35,36,37,38,39,40]. In addition, many scholars have considered the possible factor of re-positives and designed a new model based on the traditional SEIR-type to analyze the spread of the epidemic [41,42], which can also be seen in previous studies [43,44,45,46,47,48,49,50]. According to previous studies, re-positives and environmental vectors significantly impact COVID-19 dynamics and evolution, and need to be investigated widely in order to uncover the strength of disease severity and infectiousness. It has not yet been extensively explored using mathematical modeling studies. Therefore, it is necessary to use differential equation modeling methods to help understand the mechanism of environmental vectors and re-positive factors in virus transmission.

The authors of [41] consider the situation of re-infection, where recovered people may return to being considered infected people (including symptomatic and asymptomatic infected people), and these infected people can release virus particles into the environment. In order to simplify the model, we let exposed people to include lurkers and asymptomatic infected people. We assume that asymptomatic infected people are infectious, and that all infected people must undergo the incubation period.

In addition to the factors mentioned in [41], the following three types of factors are also considered: the first is that exposed individuals are still contagious, and exposed individuals can transmit viruses deposited on the surface of materials to healthy individuals through inanimate substances (public lift buttons, mail packages, etc.); the second is that the birth population is not zero, the mortality rate of different populations is also different, and recovered individuals may return to becoming susceptible individuals due to immune loss; the third is that we consider the recovered of re-positives to return to becoming exposed people.

Let the variables *S*(*t*), *E*(*t*), *I*(*t*), and *R*(*t*) represent the numbers of susceptible individuals, exposed individuals, infected individuals, and recovered individuals at time *t*, respectively, and *M*(*t*) represent the number of free virus particles accumulated in environmental vectors. The block diagram of the interactions among *S*, *E*, *I*, *R*, and *M* is shown below (Figure 1).

According to the above block diagram of the interaction between these state variables, we have the following five-dimensional ordinary differential equation model:(1)$$\left\{\begin{array}{cc} \dot{S}\left(t\right)= & \Lambda -\beta_{1}S\left(t\right)I\left(t\right)-\beta_{2}S\left(t\right)E\left(t\right)-\gamma_{3}S\left(t\right)M\left(t\right)+\mu_{4}R\left(t\right)-dS\left(t\right),\\ \dot{E}\left(t\right)= & \beta_{1}S\left(t\right)I\left(t\right)+\beta_{2}S\left(t\right)E\left(t\right)+\gamma_{3}S\left(t\right)M\left(t\right)+\mu_{3}R\left(t\right)-\left(d+d_{1}+\mu_{1}\right)E\left(t\right),\\ \dot{I}\left(t\right)= & \mu_{1}E\left(t\right)-\left(d+d_{2}+\mu_{2}\right)I\left(t\right),\\ \dot{R}\left(t\right)= & \mu_{2}I\left(t\right)-\left(d+\mu_{3}+\mu_{4}\right)R\left(t\right),\\ \dot{M}\left(t\right)= & \gamma_{1}E\left(t\right)+\gamma_{2}I\left(t\right)-\rho M\left(t\right). \end{array}\right.$$

The biological meanings of each parameter in model (1) are shown in Table 1 below:

Considering the biological meaning, all the parameters in Table 1 are non-negative constants, Λ>0, d>0, and ρ>0.

The main purpose of the paper can be divided into two parts. Firstly, the local and global stability of the equilibria and uniform persistence of model (1) are analyzed in detail by using the basic reproduction number. Secondly, model (1) is applied to data fitting of COVID-19 from Japan and Italy, which shows the effectiveness and predictability of the model. Furthermore, the paper conducts a sensitivity analysis of the parameters in model (1) using data from Italy, which allows us to identify the main factors related to disease transmission (for example, the infection rate of infected individuals ($\beta_{1}$), infection rate of exposed individuals ($\beta_{2}$), rate of virus release into the environment by infected individuals ($\gamma_{2}$), infection rate due to environmental vectors ($\gamma_{3}$), rate of decay or clearance of free virus particles in environmental vectors (ρ), rate at which exposed individuals become infected ($\mu_{1}$), recovery rate of infected people ($\mu_{2}$), rate of re-positives ($\mu_{3}$), and the rate at which recovered individuals without immune protection return to the susceptible class ($\mu_{4}$)).

## 2. Stability and Uniform Persistence

The initial condition of model (1) is
(2)$$S\left(0\right)=S^{0},\;\;E\left(0\right)=E^{0},\;\;I\left(0\right)=I^{0},\;\;R\left(0\right)=R^{0},\;\;M\left(0\right)=M^{0},$$
where $S^{0},$$E^{0},$$I^{0},$$R^{0}$, and $M^{0}$ are non-negative constants.

It is easy to prove that the solution (S(t),E(t),I(t),R(t),M(t)) of model (1) with the initial condition (2) is existent, unique, non-negative, and ultimately bounded in [0,+∞), and satisfies
$$\underset{t\to \infty }{lim sup}\left(S\left(t\right)+E\left(t\right)+I\left(t\right)+R\left(t\right)\right)\leq \Lambda d^{-1},\;\;\;\;\;\underset{t\to \infty }{lim sup}M\left(t\right)\leq \left(\gamma_{1}+\gamma_{2}\right)\Lambda {\left(d\rho \right)}^{-1}.$$

### 2.1. Basic Reproduction Number and Classification of Equilibria

Model (1) always has the disease-free equilibrium $Q_{0}=\left(S_{0},0,0,0,0\right)$, where $S_{0}=\Lambda d^{-1}$. By the method of the next generation matrix [51,52], we obtain the basic reproduction number of model (1), as follows:$$R_{0}=\rho \left(FV^{-1}\right)=\frac{\Lambda \left(d+\mu_{3}+\mu_{4}\right)\left[\left(\rho \beta_{2}+\gamma_{1}\gamma_{3}\right)\left(d+d_{2}+\mu_{2}\right)+\mu_{1}\left(\rho \beta_{1}+\gamma_{2}\gamma_{3}\right)\right]}{d\rho \left[\left(d+d_{1}+\mu_{1}\right)\left(d+d_{2}+\mu_{2}\right)\left(d+\mu_{3}+\mu_{4}\right)-\mu_{1}\mu_{2}\mu_{3}\right]}.$$
where
$$\begin{array}{cc} & F=\left[\begin{array}{cccc} \frac{\beta_{2}\Lambda }{d} & \frac{\beta_{1}\Lambda }{d} & 0 & \frac{\gamma_{3}\Lambda }{d}\\ 0 & 0 & 0 & 0\\ 0 & 0 & 0 & 0\\ 0 & 0 & 0 & 0 \end{array}\right],\;V=\left[\begin{array}{cccc} d+d_{1}+\mu_{1} & 0 & -\mu_{3} & 0\\ -\mu_{1} & d+d_{2}+\mu_{2} & 0 & 0\\ 0 & -\mu_{2} & d+\mu_{3}+\mu_{4} & 0\\ -\gamma_{1} & -\gamma_{2} & 0 & \rho \end{array}\right]. \end{array}$$

It is not difficult to see that $R_{0}$ can be written as $R_{0}=R_{1}+R_{2}+R_{3}$, where
$$\begin{array}{cc} & R_{1}=\frac{\mu_{1}\left(d+\mu_{3}+\mu_{4}\right)\beta_{1}S_{0}}{\left(d+d_{1}+\mu_{1}\right)\left(d+d_{2}+\mu_{2}\right)\left(d+\mu_{3}+\mu_{4}\right)-\mu_{1}\mu_{2}\mu_{3}},\\ & R_{2}=\frac{\left(d+d_{2}+\mu_{2}\right)\left(d+\mu_{3}+\mu_{4}\right)\beta_{2}S_{0}}{\left(d+d_{1}+\mu_{1}\right)\left(d+d_{2}+\mu_{2}\right)\left(d+\mu_{3}+\mu_{4}\right)-\mu_{1}\mu_{2}\mu_{3}},\\ & R_{3}=\frac{\left[\mu_{1}\gamma_{2}+\gamma_{1}\left(d+d_{2}+\mu_{2}\right)\right]\left(d+\mu_{3}+\mu_{4}\right)\gamma_{3}S_{0}}{\rho \left[\left(d+d_{1}+\mu_{1}\right)\left(d+d_{2}+\mu_{2}\right)\left(d+\mu_{3}+\mu_{4}\right)-\mu_{1}\mu_{2}\mu_{3}\right]}. \end{array}$$

According to the expressions of $R_{1}$, $R_{2}$, and $R_{3}$, it can be clearly seen that the value of the basic reproduction number $R_{0}$ directly depends on the infection rate of the infected individuals ($\beta_{1}$), the infection rate of exposed individuals ($\beta_{2}$), and the infection rate due to environmental vectors ($\gamma_{3}$).

Assuming that (S,E,I,R,M) can be any equilibrium of model (1), it has the following equations:(3)$$\left\{\begin{array}{c} \Lambda -\beta_{1}SI-\beta_{2}SE-\gamma_{3}SM+\mu_{4}R-dS=0,\\ \beta_{1}SI+\beta_{2}SE+\gamma_{3}SM+\mu_{3}R-\left(d+d_{1}+\mu_{1}\right)E=0,\\ \mu_{1}E-\left(d+d_{2}+\mu_{2}\right)I=0,\\ \mu_{2}I-\left(d+\mu_{3}+\mu_{4}\right)R=0,\\ \gamma_{1}E+\gamma_{2}I-\rho M=0. \end{array}\right.$$

By solving the equations (3), it is easy to obtain that when $R_{0}>1$, model (1) has a unique endemic equilibrium $Q^{*}=\left(S^{*},E^{*},I^{*},R^{*},M^{*}\right)$, where
$$\begin{array}{cc} S^{*} & =\frac{\rho \left(d+d_{1}+\mu_{1}\right)\left(d+d_{2}+\mu_{2}\right)\left(d+\mu_{3}+\mu_{4}\right)-\rho \mu_{1}\mu_{2}\mu_{3}}{\mu_{1}\left(\rho \beta_{1}+\gamma_{2}\gamma_{3}\right)\left(d+\mu_{3}+\mu_{4}\right)+\left(\rho \beta_{2}+\gamma_{1}\gamma_{3}\right)\left(d+d_{2}+\mu_{2}\right)\left(d+\mu_{3}+\mu_{4}\right)}=\frac{\Lambda }{R_{0}d},\\ E^{*} & =\frac{\left(d+d_{2}+\mu_{2}\right)\left(d+\mu_{3}+\mu_{4}\right)}{\mu_{1}\mu_{2}}R^{*},\;\;\;\;\;I^{*}=\frac{\left(d+\mu_{3}+\mu_{4}\right)}{\mu_{2}}R^{*},\\ R^{*} & =\left(1-\frac{1}{R_{0}}\right)\frac{\mu_{1}\mu_{2}\Lambda }{\left(d+d_{1}+\mu_{1}\right)\left(d+d_{2}+\mu_{2}\right)\left(d+\mu_{3}+\mu_{4}\right)-\left(\mu_{1}\mu_{2}\mu_{3}+\mu_{1}\mu_{2}\mu_{4}\right)},\\ M^{*} & =\frac{\gamma_{1}\left(d+d_{2}+\mu_{2}\right)\left(d+\mu_{3}+\mu_{4}\right)+\mu_{1}\gamma_{2}\left(d+\mu_{3}+\mu_{4}\right)}{\mu_{1}\mu_{2}\rho }R^{*}. \end{array}$$

### 2.2. Global Stability of the Disease-Free Equilibrium

It is clear that the set
$$\Omega =\left\{\left(E,I,R,M\right)\;|\;E,I,R,M\geq 0,\;S+E+I+R\leq S_{0},\;M\leq \left(\gamma_{1}+\gamma_{2}\right)\Lambda {\left(d\rho \right)}^{-1}\right\}$$
is attractive and positively invariant with respect to model (1). Thus, this leads to the following theorem:

**Theorem 1.** *If $R_{0}<1$, the disease-free equilibrium $Q_{0}$ of model* (1) *is globally asymptotically stable with respect to* Ω.

**Proof.** Firstly, let us show that the disease-free equilibrium $Q_{0}$ is locally asymptotically stable. For convenience, let $d+d_{1}+\mu_{1}=a_{1},\;d+d_{2}+\mu_{2}=a_{2}$, and $\;d+\mu_{3}+\mu_{4}=a_{3}$. Hence, $a_{1}a_{2}a_{3}-\mu_{1}\mu_{2}\mu_{3}>0$.Using simple calculations, we can determine that the characteristic equation of model (1) at the disease-free equilibrium $Q_{0}$ is (λ+d)g(λ)=0, where
$$\begin{array}{cc} g(\lambda )=\; & -\frac{\gamma_{3}\Lambda }{d}\left(\lambda +a_{3}\right)\left[\mu_{1}\gamma_{2}+\gamma_{1}\left(\lambda +a_{2}\right)\right]+\left(\lambda +\rho \right)\left(\lambda -\frac{\beta_{2}\Lambda }{d}+a_{1}\right)\left(\lambda +a_{2}\right)\left(\lambda +a_{3}\right)\\ \; & -\left(\lambda +\rho \right)[\frac{\mu_{1}\beta_{1}\Lambda }{d}\left(\lambda +a_{3}\right)+\mu_{1}\mu_{2}\mu_{3}]. \end{array}$$Clearly, $\lambda_{1}=-d$ is a negative root. Let us further show that when $R_{0}<1$, all the roots of the equation g(λ)=0 have negative real parts. Suppose that λ=x+iy(x≥0) is an eigenvalue of the equation g(λ)=0.Then, the equation g(λ)=0 can be rewritten in the following equivalent form:
$$\begin{array}{ccc} 1= & \frac{\mu_{1}\gamma_{2}\gamma_{3}\Lambda }{d\left(\lambda +\rho \right)\left(\lambda +a_{1}\right)\left(\lambda +a_{2}\right)}+\frac{\gamma_{1}\gamma_{3}\Lambda }{d\left(\lambda +\rho \right)\left(\lambda +a_{1}\right)}+\frac{\beta_{2}\Lambda }{d(\lambda +a_{1})}\; & +\frac{\mu_{1}\beta_{1}\Lambda }{d\left(\lambda +a_{1}\right)\left(\lambda +a_{2}\right)}\\ \; & +\frac{\mu_{1}\mu_{2}\mu_{3}}{\left(\lambda +a_{1}\right)\left(\lambda +a_{2}\right)\left(\lambda +a_{3}\right)}. \end{array}$$Taking the modulus on both sides of the above equality, it becomes
$$\begin{array}{c} 1=|\frac{\mu_{1}\gamma_{2}\gamma_{3}\Lambda }{d\left(\lambda +\rho \right)\left(\lambda +a_{1}\right)\left(\lambda +a_{2}\right)}+\frac{\gamma_{1}\gamma_{3}\Lambda }{d\left(\lambda +\rho \right)\left(\lambda +a_{1}\right)}+\frac{\beta_{2}\Lambda }{d(\lambda +a_{1})}+\frac{\mu_{1}\beta_{1}\Lambda }{d\left(\lambda +a_{1}\right)\left(\lambda +a_{2}\right)}\\ \;\;\;\;\;+\frac{\mu_{1}\mu_{2}\mu_{3}}{\left(\lambda +a_{1}\right)\left(\lambda +a_{2}\right)\left(\lambda +a_{3}\right)}|\\ \;\leq |\frac{\mu_{1}\gamma_{2}\gamma_{3}\Lambda }{d\left(\lambda +\rho \right)\left(\lambda +a_{1}\right)\left(\lambda +a_{2}\right)}\left|+\right|\frac{\gamma_{1}\gamma_{3}\Lambda }{d\left(\lambda +\rho \right)\left(\lambda +a_{1}\right)}\left|+\right|\frac{\beta_{2}\Lambda }{d(\lambda +a_{1})}\left|+\right|\frac{\mu_{1}\beta_{1}\Lambda }{d\left(\lambda +a_{1}\right)\left(\lambda +a_{2}\right)}|\\ \;\;\;\;+|\frac{\mu_{1}\mu_{2}\mu_{3}}{\left(\lambda +a_{1}\right)\left(\lambda +a_{2}\right)\left(\lambda +a_{3}\right)}|\\ \;\leq \frac{\mu_{1}\gamma_{2}\gamma_{3}\Lambda }{d\rho a_{1}a_{2}}+\frac{\gamma_{1}\gamma_{3}a_{2}\Lambda }{d\rho a_{1}a_{2}}+\frac{\beta_{2}\rho a_{2}\Lambda }{d\rho a_{1}a_{2}}+\frac{\mu_{1}\beta_{1}\rho \Lambda }{d\rho a_{1}a_{2}}+\frac{\mu_{1}\mu_{2}\mu_{3}}{a_{1}a_{2}a_{3}}\\ \;=\frac{\Lambda (\mu_{1}\gamma_{2}\gamma_{3}+\mu_{1}\beta_{1}\rho +\gamma_{1}\gamma_{3}a_{2}+\beta_{2}\rho a_{2})}{d\rho a_{1}a_{2}}+\frac{\mu_{1}\mu_{2}\mu_{3}}{a_{1}a_{2}a_{3}}\\ \;=\frac{\left(a_{1}a_{2}a_{3}-\mu_{1}\mu_{2}\mu_{3}\right)}{a_{1}a_{2}a_{3}}R_{0}+\frac{\mu_{1}\mu_{2}\mu_{3}}{a_{1}a_{2}a_{3}}\\ \;=R_{0}+\left(1-R_{0}\right)\frac{\mu_{1}\mu_{2}\mu_{3}}{a_{1}a_{2}a_{3}}. \end{array}$$Note that $a_{1}a_{2}a_{3}-\mu_{1}\mu_{2}\mu_{3}>0$ and $R_{0}<1$, meaning that the above inequality is not valid. This proves that when $R_{0}<1$, all the roots of the equation g(λ)=0 have negative real parts. Hence, the disease-free equilibrium $Q_{0}$ is locally asymptotically stable.Next, we show that the disease-free equilibrium $Q_{0}$ is globally attractable. In fact, for t≥0, model (1) implies
$$\left\{\begin{array}{c} \dot{E}\left(t\right)\leq \beta_{1}S_{0}I\left(t\right)+\beta_{2}S_{0}E\left(t\right)+\gamma_{3}S_{0}M\left(t\right)+\mu_{3}R\left(t\right)-\left(d+d_{1}+\mu_{1}\right)E\left(t\right),\\ \dot{I}\left(t\right)\leq \mu_{1}E\left(t\right)-\left(d+d_{2}+\mu_{2}\right)I\left(t\right),\\ \dot{R}\left(t\right)\leq \mu_{2}I\left(t\right)-\left(d+\mu_{3}+\mu_{4}\right)R\left(t\right),\\ \dot{M}\left(t\right)\leq \gamma_{1}E\left(t\right)+\gamma_{2}I\left(t\right)-\rho M\left(t\right). \end{array}\right.$$Let $Y={\left(y_{1},y_{2},y_{3},y_{4}\right)}^{T}$, and consider the comparison system $\frac{dY}{dt}=\left(F-V\right)Y$. As the condition $R_{0}=\rho \left(FV^{-1}\right)<1$ means that all eigenvalues of matrix F−V have negative real parts, the trivial solution of the comparison system is asymptotically stable. Therefore, it follows that $E\left(t\right)\leq y_{1}\left(t\right)\to 0$, $I\left(t\right)\leq y_{2}\left(t\right)\to 0$, $R\left(t\right)\leq y_{3}\left(t\right)\to 0$ and $M\left(t\right)\leq y_{4}\left(t\right)\to 0$ for t→+∞. Furthermore, from the first equation of model (Equation 1), it is easy to obtain $S\left(t\right)\to S_{0}$ for t→+∞. This proves that the disease-free equilibrium $Q_{0}$ is globally attractive with respect to the set Ω. □

### 2.3. Uniform Persistence and Global Stability of the Endemic Equilibrium

In the subsection, there is the following Theorem 2 for the uniform persistence of model (1).

**Theorem 2.** *If $R_{0}>1$, then model* (1) *is uniformly persistent, and each positive solution ${\left(S\left(t\right),\;E\left(t\right),\;I\left(t\right),\;R\left(t\right),\;M\left(t\right)\right)}^{T}$ of model* (1) *with the initial condition* (2) *satisfies*
$$\begin{array}{cc} & \underset{t\to \infty }{lim inf}S\left(t\right)\geq \frac{\Lambda d}{\Lambda \left[\beta_{1}+\beta_{2}+\gamma_{3}\left(\gamma_{1}+\gamma_{2}\right)\rho^{-1}\right]+d^{2}}\equiv v_{1},\;\;\;\;\;\underset{t\to \infty }{lim inf}E\left(t\right)\geq \delta E^{*}e^{-Ta_{1}}\equiv v_{2},\\ & \underset{t\to \infty }{lim inf}I\left(t\right)\geq \frac{\mu_{1}}{a_{2}}v_{2},\;\;\;\;\underset{t\to \infty }{lim inf}R\left(t\right)\geq \frac{\mu_{1}\mu_{2}}{a_{2}a_{3}}v_{2},\;\;\;\;\;\underset{t\to \infty }{lim inf}M\left(t\right)\geq \left(\frac{\gamma_{1}}{\rho }+\frac{\gamma_{2}\mu_{1}}{a_{2}\rho }\right)v_{2}, \end{array}$$
*where δ>0 and T>0, and satisfies*
$$q\equiv \left(\frac{\beta_{1}v_{2}}{a_{2}}+\beta_{2}+\frac{\gamma_{1}\gamma_{3}}{\rho }+\frac{\gamma_{2}\gamma_{3}\mu_{1}}{a_{2}\rho }\right)\delta E^{*}+d,\;\frac{\Lambda }{q}>S^{*},\;S^{\Delta }\equiv \frac{\Lambda }{q}\left(1-e^{-Tq}\right)>S^{*}.$$

To complete the proof of Theorem 2, the key point is to show that the estimation $\underset{t\to +\infty }{lim inf}E\left(t\right)\geq v_{2}$ holds. The detailed proof is similar to [53,54,55,56] and has been omitted here.

From a biological point of view, Theorem 2 indicates that as long as the basic reproduction number $R_{0}>1$, the disease infection cannot be eliminated and will permanently exist.

Next, let us consider the global asymptotic stability of the endemic equilibrium $Q^{*}$ of model (1). To simplify model (1), it is assumed that the death rate of the exposed and infected individuals caused by the virus is zero, i.e., (H1)$d_{1}=d_{2}=0$.

Let N(t)=S(t)+E(t)+I(t)+R(t), then for t≥0, we have $\dot{N}\left(t\right)=\Lambda -dN\left(t\right)$, which implies $\lim_{t\to +\infty }N\left(t\right)=S_{0}$. On the hyperplane $N=S_{0}$, model (1) can be transformed into the following equivalent four-dimensional system:(4)$$\left\{\begin{array}{cc} \dot{S}\left(t\right)= & \Lambda -\beta_{1}S\left(t\right)I\left(t\right)-\beta_{2}S\left(t\right)E\left(t\right)-\gamma_{3}S\left(t\right)M\left(t\right)+\mu_{4}\left(S_{0}-S\left(t\right)-E\left(t\right)-I\left(t\right)\right)-dS\left(t\right),\\ \dot{E}\left(t\right)= & \beta_{1}S\left(t\right)I\left(t\right)\;+\;\beta_{2}S\left(t\right)E\left(t\right)\;+\;\gamma_{3}S\left(t\right)M\left(t\right)\;+\;\mu_{3}\left(S_{0}\;-\;S\left(t\right)\;-\;E\left(t\right)\;-\;I\left(t\right)\right)\;-\;\left(d\;+\;\mu_{1}\right)E\left(t\right),\\ \dot{I}\left(t\right)= & \mu_{1}E\left(t\right)-\left(d+\mu_{2}\right)I\left(t\right),\\ \dot{M}\left(t\right)= & \gamma_{1}E\left(t\right)+\gamma_{2}I\left(t\right)-\rho M\left(t\right). \end{array}\right.$$

In addition, the following conditions are also used:

(H2)$\mu_{2}+\mu_{3}-2\mu_{1}-\mu_{4}>0,\;\;d+\mu_{1}+\mu_{3}-\left(2\beta_{1}+2\beta_{2}\right)S_{0}-\rho -2\mu_{4}>0,\;\;d+\mu_{2}-\mu_{1}>0$, $\gamma_{1}\geq \gamma_{2}$, $2\mu_{4}\geq \mu_{1}+\mu_{3}.$

By using a similar method to what was used in [57,58,59], we have Theorem 3.

**Theorem 3.** *If the conditions (H1)–(H2) hold, then, when $R_{0}>1$, the endemic equilibrium $Q^{*}$ of system* (4) *is globally asymptotically stable with respect to* Ω.

**Proof.** For system (4), the Jacobian matrix *J* and second additive compound matrix $J^{\left[2\right]}$ are given by
$$J=\left(\begin{array}{cccc} -\beta_{1}I-\beta_{2}E-\gamma_{3}M-d-\mu_{4} & -\beta_{2}S-\mu_{4} & -\beta_{1}S-\mu_{4} & -\gamma_{3}S\\ \beta_{1}I+\beta_{2}E+\gamma_{3}M-\mu_{3} & \beta_{2}S-d-\mu_{1}-\mu_{3} & \beta_{1}S-\mu_{3} & \gamma_{3}S\\ 0 & \mu_{1} & -d-\mu_{2} & 0\\ 0 & \gamma_{1} & \gamma_{2} & -\rho \end{array}\right)$$
and
$$J^{\left[2\right]}=\left(\begin{array}{cccccc} M_{11} & \beta_{1}S-\mu_{3} & \gamma_{3}S & \beta_{1}S+\mu_{4} & \gamma_{3}S & 0\\ \mu_{1} & M_{22} & 0 & -\beta_{2}S-\mu_{4} & 0 & \gamma_{3}S\\ \gamma_{1} & \gamma_{2} & M_{33} & 0 & -\beta_{2}S-\mu_{4} & -\beta_{1}S-\mu_{4}\\ 0 & q & 0 & M_{44} & 0 & -\gamma_{3}S\\ 0 & 0 & q & \gamma_{2} & M_{55} & \beta_{1}S-\mu_{3}\\ 0 & 0 & 0 & -\gamma_{1} & \mu_{1} & M_{66} \end{array}\right),$$
where$\;M_{11}=-\beta_{1}I-\beta_{2}E-\gamma_{3}M-d-\mu_{4}+\beta_{2}S-d-\mu_{1}-\mu_{3}$,$\;M_{22}=-\beta_{1}I-\beta_{2}E-\gamma_{3}M-d-\mu_{4}-d-\mu_{2}$,$\;M_{33}=-\beta_{1}I-\beta_{2}E-\gamma_{3}M-d-\mu_{4}-\rho$,$\;M_{44}=\beta_{2}S-2d-\mu_{1}-\mu_{3}-\mu_{2}$,$\;M_{55}=\beta_{2}S-d-\mu_{1}-\mu_{3}-\rho$,$\;M_{66}=-d-\mu_{2}-\rho$,$\;q=\beta_{1}I+\beta_{2}E+\gamma_{3}M-\mu_{3}$.Let
$$P=P\left(S,E,I,M\right)=\left(\begin{array}{cccccc} \frac{1}{E} & 0 & 0 & 0 & 0 & 0\\ 0 & \frac{1}{E} & 0 & 0 & 0 & 0\\ 0 & 0 & 0 & \frac{1}{E} & 0 & 0\\ 0 & 0 & \frac{1}{M} & 0 & 0 & 0\\ 0 & 0 & 0 & 0 & \frac{1}{M} & 0\\ 0 & 0 & 0 & 0 & 0 & \frac{1}{M} \end{array}\right),$$
using *f* to represent the vector form of system (4). Then, $P_{f}$ is the directional derivative of P(x) along the direction of *f*. It follows that $P_{f}P^{-1}=diag\left\{-\frac{\dot{E}}{E},-\frac{\dot{E}}{E},-\frac{\dot{E}}{E},-\frac{\dot{M}}{M},-\frac{\dot{M}}{M},-\frac{\dot{M}}{M}\right\}$, and
$$\begin{array}{cc} Q(S,E,I,M)= & P_{f}P^{-1}+PJ^{\left[2\right]}P^{-1}\\ = & \left(\begin{array}{cccccc} M_{11}-\frac{\dot{E}}{E} & \beta_{1}S-\mu_{3} & \beta_{1}S+\mu_{4}\; & \frac{\gamma_{3}SM}{E} & \frac{\gamma_{3}SM}{E} & 0\\ \mu_{1} & M_{22}-\frac{\dot{E}}{E} & -\beta_{2}S-\mu_{4} & 0 & 0 & \frac{\gamma_{3}SM}{E}\\ 0 & q & M_{44}-\frac{\dot{E}}{E} & 0 & 0 & -\frac{\gamma_{3}SM}{E}\\ \frac{\gamma_{2}E}{M} & \frac{\gamma_{2}E}{M} & 0 & M_{33}-\frac{\dot{M}}{M} & -\beta_{2}S-\mu_{4} & -\beta_{1}S-\mu_{4}\\ 0 & 0 & \frac{\gamma_{2}E}{M} & q & M_{55}-\frac{\dot{M}}{M} & \beta_{1}S-\mu_{3}\\ 0 & 0 & -\frac{\gamma_{1}E}{M} & 0 & \mu_{1} & M_{55}-\frac{\dot{M}}{M} \end{array}\right). \end{array}$$The matrix $Q\left(S,E,I,M\right)$ can be written in block form: $Q\left(S,E,I,M\right)={\left(Q_{ij}\right)}_{4\times 4}$, where
$$\begin{array}{cc} \; & Q_{11}=M_{11}-\frac{\dot{E}}{E},\;Q_{12}=\left(\beta_{1}S-\mu_{3},\beta_{1}S+\mu_{4}\right),\;Q_{13}=\left(\frac{\gamma_{3}SM}{E},\frac{\gamma_{3}SM}{E}\right),\;Q_{14}=0,\\ \; & Q_{21}={\left(\mu_{1},0\right)}^{T},\;Q_{22}=\left(\begin{array}{cc} M_{22}-\frac{\dot{E}}{E} & -\beta_{2}S-\mu_{4}\\ q & M_{44}-\frac{\dot{E}}{E} \end{array}\right),\;Q_{23}=\left(\begin{array}{cc} 0 & 0\\ 0 & 0 \end{array}\right),\\ \; & Q_{24}={\left(\frac{\gamma_{3}SM}{E},-\frac{\gamma_{3}SM}{E}\right)}^{T},\;Q_{31}={\left(\frac{\gamma_{2}E}{M},0\right)}^{T},\;Q_{32}=\left(\begin{array}{cc} \frac{\gamma_{2}E}{M} & 0\\ 0 & \frac{\gamma_{2}E}{M} \end{array}\right),\\ \; & Q_{33}=\left(\begin{array}{cc} M_{33}-\frac{\dot{M}}{M} & -\beta_{2}S-\mu_{4}\\ q & M_{55}-\frac{\dot{M}}{M} \end{array}\right),\;Q_{34}={\left(-\beta_{1}S-\mu_{4},\beta_{1}S-\mu_{3}\right)}^{T},\;Q_{41}=0,\\ \; & Q_{42}=\left(0,-\frac{\gamma_{1}E}{M}\right),\;Q_{43}=\left(0,\mu_{1}\right),\;Q_{44}=M_{66}-\frac{\dot{M}}{M}. \end{array}$$Hence, we have the following estimation (for an example, see [57,58,59]),
$$\sigma \left(Q\left(S,E,I,M\right)\right)\leq \sup\left\{g_{1},g_{2},g_{3},g_{4}\right\},$$
where $g_{1}=\sigma_{1}\left(Q_{11}\right)+\left|Q_{12}\right|+\left|Q_{13}\right|+\left|Q_{14}\right|,$ $g_{2}=\sigma_{1}\left(Q_{22}\right)+\left|Q_{21}\right|+\left|Q_{23}\right|+\left|Q_{24}\right|,$ $g_{3}=\sigma_{1}\left(Q_{33}\right)+\left|Q_{31}\right|+\left|Q_{32}\right|+\left|Q_{34}\right|,$$g_{4}=\sigma_{1}\left(Q_{44}\right)+\left|Q_{41}\right|+\left|Q_{42}\right|+\left|Q_{43}\right|.$ $\left|Q_{ij}\right|\left(i\neq j,i,j=1,2,3,4\right)$ are matrix norms with respect to the $l_{1}$ vector norm, and $\sigma_{1}$ denotes the Lozinski $\overset{˘}{ı}$ measures with respect to the $l_{1}$ norm. Furthermore, it is easy to obtain that
$$\begin{array}{cc} \sigma (Q_{11})= & -\beta_{1}I-\beta_{2}E-\gamma_{3}M-2d-\mu_{4}+\beta_{2}S-\mu_{1}-\mu_{3}-\frac{\dot{E}}{E},\;\;\;\;\;\;\;\;\;\;\;\;\\ \sigma (Q_{22})\leq & -\mu_{2}-2d-\frac{\dot{E}}{E}+\max\left\{-\mu_{4},2\beta_{2}S+\mu_{4}-\mu_{1}-\mu_{3}\right\},\;\;\;\\ \sigma (Q_{33})\leq & -\rho -d-\frac{\dot{M}}{M}+\max\left\{-\mu_{4},2\beta_{2}S+\mu_{4}-\mu_{1}-\mu_{3}\right\},\;\;\;\\ \sigma (Q_{44})= & -d-\mu_{2}-\rho -\frac{\dot{M}}{M},\;\;\;\;\;\left|Q_{12}\right|=\beta_{1}S+\mu_{4},\;\left|Q_{13}\right|=\frac{\gamma_{3}SM}{E}, \end{array}$$
$$\left|Q_{14}\right|=0,\;\left|Q_{21}\right|=\mu_{1},\;\left|Q_{23}\right|=0,\;\left|Q_{24}\right|=\frac{2\gamma_{3}SM}{E},\;\left|Q_{31}\right|=\frac{\gamma_{2}E}{M},$$
$$\left|Q_{32}\right|=\frac{\gamma_{2}E}{M},\;\left|Q_{34}\right|<2\beta_{1}S+\mu_{4},\;\left|Q_{41}\right|=0,\;\left|Q_{42}\right|=\frac{\gamma_{1}E}{M},\;\left|Q_{43}\right|=\mu_{1}.$$From condition (H2) and system (4), we have
$$\max\left\{-\mu_{4},2\beta_{2}S+\mu_{4}-\mu_{1}-\mu_{3}\right\}=2\beta_{2}S+\mu_{4}-\mu_{1}-\mu_{3},$$
and
$$\begin{array}{cc} \frac{\dot{E}}{E}= & \frac{\beta_{1}SI}{E}+\beta_{2}S+\frac{\gamma_{3}SM}{E}+\frac{\mu_{3}R}{E}-\left(d+\mu_{1}\right)>\frac{\beta_{1}SI}{E}+\beta_{2}S+\frac{\gamma_{3}SM}{E}-\left(d+\mu_{1}\right),\\ \frac{\dot{M}}{M}= & \frac{\gamma_{1}E}{M}+\frac{\gamma_{2}I}{M}-\rho >\frac{\gamma_{1}E}{M}-\rho \geq \frac{\gamma_{2}E}{M}-\rho . \end{array}$$Hence, we have
$$\begin{array}{cc} g_{1}= & -\beta_{1}I-\beta_{2}E-\gamma_{3}M-2d-\mu_{4}+\beta_{2}S-\mu_{1}-\mu_{3}-\frac{\dot{E}}{E}+\beta_{1}S+\mu_{4}+\frac{\gamma_{3}SM}{E}\\ < & -\beta_{1}I-\beta_{2}S-\gamma_{3}M-\frac{\beta_{1}SI}{E}-d-\mu_{3}+\beta_{1}S\\ < & \frac{\beta_{1}\Lambda }{d}-d-\mu_{3}\equiv -\theta_{1},\\ g_{2}\leq & -\mu_{2}-2d-\mu_{3}-\frac{\dot{E}}{E}+2\beta_{2}S+\mu_{4}-\mu_{1}+\mu_{1}+\frac{2\gamma_{3}SM}{E}\\ < & -\mu_{2}-d-\mu_{3}+\frac{\gamma_{3}SM}{E}+\beta_{2}S-\frac{\beta_{1}SI}{E}+\mu_{1}+\mu_{4}\\ < & \frac{\dot{E}}{E}+2\mu_{1}+\mu_{4}-\mu_{2}-\mu_{3}\equiv \frac{\dot{E}}{E}-\theta_{2},\\ g_{3}< & -\rho -\mu_{3}-d-\frac{\dot{M}}{M}+2\beta_{2}S+\mu_{4}-\mu_{1}+\frac{2\gamma_{2}E}{M}+2\beta_{1}S+\mu_{4}\\ < & \frac{\gamma_{2}E}{M}-\rho +\rho -\mu_{3}+\left(2\beta_{1}+2\beta_{2}\right)S+2\mu_{4}-d-\mu_{1}\\ < & \frac{\dot{M}}{M}+\left(2\beta_{1}+2\beta_{2}\right)\frac{\Lambda }{d}+\rho +2\mu_{4}-d-\mu_{1}-\mu_{3}\equiv \frac{\dot{M}}{M}-\theta_{3},\\ g_{4}= & -d-\mu_{2}-\rho -\frac{\dot{M}}{M}+\frac{\gamma_{1}E}{M}+\mu_{1}\\ < & \mu_{1}-d-\mu_{2}\equiv -\theta_{4}. \end{array}$$Again, from condition (H2), we can obtain
$$\theta_{3}<\theta_{1},\;\bar{b}=\min\left\{\theta_{1},\theta_{2},\theta_{3},\theta_{4}\right\}=\min\left\{\theta_{2},\theta_{3},\theta_{4}\right\}>0,$$
and
$$g_{1}\leq -\bar{b},\;g_{2}<\frac{\dot{E}}{E}-\bar{b},\;g_{3}<\frac{\dot{M}}{M}-\bar{b},\;g_{4}\leq -\bar{b}.$$Note that when $R_{0}>1$, system (4) is uniformly persistent. Using a similar argument as in [59], we find that the endemic equilibrium $Q^{*}$ of system (4) is globally asymptotically stable. □

Take Λ=1000, $\beta_{1}=5\times 10^{-7}$, $\beta_{2}=5\times 10^{-7}$, $\gamma_{1}=2\times 10^{-3}$, $\gamma_{2}=2\times 10^{-3}$, $\gamma_{3}=2\times 10^{-3}$, $\mu_{1}=0.2$, $\mu_{2}=0.5$, $\mu_{3}=0.1$, $\mu_{4}=0.15$, d=0.1, $d_{1}=0.1,\;\;d_{2}=0.1,$ and ρ=0.05. Using numerical calculations, we obtain the basic reproduction number $R_{0}=2.625>1$, and the inequalities in (H1)–(H2) hold.

## 3. Applications of the Model in Japan and Italy

In this section, we use model (1) to fit the data of confirmed cases and recovered cases of COVID-19 in Japan and Italy (data were taken from the Johns Hopkins University Center for Systems Science and Engineering, https://github.com/CSSEGISandData/COVID-19 accessed on 1 July 2022), and make short-term predictions about disease infection trends. Meanwhile, based on the basic reproduction number $R_{0}$ and data in Italy, we carried out a sensitivity analysis and the main factors related to disease transmission were captured.

### 3.1. Predicted Cases for Cumulative Confirmed and Recovered Cases Based on Data in Japan and Italy

First, from [3,60,61,62] and data published by the World Health Organization (WHO), the parameters $\frac{1}{\mu_{1}}$ (incubation period), $\frac{1}{\mu_{2}}$ (infectious period), $\frac{1}{\mu_{4}}$ (time of immune protection), $\frac{1}{d}$ (mean lifetime), and $d_{2}$ (death rate of the infected individuals caused by the virus) in model (1) can take the values as shown in Table 2 below. Furthermore, by using data of COVID-19 in Japan (20 May–18 June 2022) and Italy (12 January–10 February 2022) and the least squares method (LSM), the remaining parameters in model (1) take values as shown in Table 2 below.

Based on the parameter values given in Table 2, we obtain Figure 2a,b, which shows that model (1) fits well with the evolution of cumulative confirmed cases and recovered cases in Japan from 20 May–18 June 2022. Similarly, we have Figure 3a,b that shows that model (1) also fits well with the evolution of the cumulative confirmed cases and recovered cases in Italy from 12 January–10 February 2022.

Next, we use model (1) and the parameter values given in Table 2 to predict the cumulative confirmed cases and recovered cases in Japan (from 19–28 June 2022) and Italy (from 11–20 February 2022). These are shown in Table 3, Table 4, Table 5 and Table 6.

From Table 3, Table 4, Table 5 and Table 6, it can be seen that the predicted values for confirmed and recovered cases are within a range of 97 to 103% of the reported data. With the increase in time *t*, the relative error also increases.

Furthermore, we use MAPE (mean absolute percentage error) and RMSPE (root mean square error) to assess the reliability of model (1) in data fitting (for examples, see [63,64]). For convenience, we introduce the definitions of MAPE and RMSPE and their evaluation standard, as follows (Table 7):$$\mathrm{MAPE}=\frac{\sum_{i=1}^{n}\left|\frac{y_{i}-y_{i}^{\prime }}{y_{i}}\right|}{n}\times 100\%,\;\;\;\;\mathrm{RMSPE}=\sqrt{\frac{\sum_{i=1}^{n}{\left[\frac{y_{i}-y_{i}^{\prime }}{y_{i}}\right]}^{2}}{n-1}}\times 100\%,$$

Here, $y_{i}$ and $y_{i}^{\prime }$ are the reported data and predicted data, respectively, and the positive integer *n* is the number of predicted data. Using the data in Table 3, Table 4, Table 5 and Table 6, we obtain Table 8.

### 3.2. Sensitivity Analysis Based on the Basic Reproduction Number and Data from Italy

From Theorem 1 and 2 we see that if the basic reproduction number $R_{0}<1$ (or $R_{0}>1$), the disease infection can be cleared (or will become an endemic disease). Therefore, it is very necessary to make a sensitivity analysis based on the basic reproduction number $R_{0}$, and then capture the main factors related to disease transmission.

The normalized sensitivity index of $R_{0}$ is defined as $\gamma_{\epsilon }^{R_{0}}\equiv \frac{\partial R_{0}}{\partial \epsilon }\frac{\epsilon }{R_{0}}.$ Here, ϵ can be regarded as any parameter in $R_{0}$ [65]. By using Italy’s parameter values in Table 3, we obtain Table 9 and Figure 4.

From Table 9 and Figure 4, we first observe that the parameters $\beta_{1}$ (the infection rate of infected individuals) and $\beta_{2}$ (the infection rate of exposed individuals) are the most sensitive (and positively correlated) parameters to the basic reproduction number $R_{0}$. When $\beta_{1}$ increases by 10% while other parameters remain unchanged, $R_{0}$ increases by 5.827%. Similarly, when $\beta_{2}$ increases by 10% while other parameters remain unchanged, $R_{0}$ increases by 2.993%. Moreover, the parameters $\mu_{1}$ (the rate at which exposed individuals become infected) and t$\mu_{2}$ (the recovery rate of infected people) also have high sensitivities (but are negatively correlated) to the basic reproduction number $R_{0}$. When $\mu_{1}$ increases by 10% while other parameters remain unchanged, $R_{0}$ decreases by 2.501%. Similarly, when $\mu_{2}$ increases by 10% while other parameters remain unchanged, $R_{0}$ decreases by 1.816%.

On the other hand, we also see from Table 9 and Figure 4 that the basic reproduction number $R_{0}$ has a strong dependence (positively/negatively correlated) on environmental and re-positive factors. Specifically, the parameters $\mu_{3}$ (the rate of re-positives), $\gamma_{2}$ (the rate of virus release into the environment by infected individuals), $\gamma_{3}$ (the infection rate due to environmental vectors), and ρ (the rate at which the free virus particles decay or are cleared in environmental vectors) are also more sensitive to the basic reproduction number $R_{0}$. When $\gamma_{2}$, $\gamma_{3}$, and $\mu_{3}$ increases by 10% while other parameters remain unchanged, $R_{0}$ increases by 0.979, 1.180 and 2.618%, respectively; however, when ρ increases by 10%, $R_{0}$ decreases by 1.180%.

This suggests that we should strengthen prevention and control to reduce the transmission rate of the disease, shorten the treatment cycle, increase antibodies by vaccination, pay attention to environmental sanitation, and reduce the risk of environmental infection.

At the end of this subsection, we present a discussion of the control strategies related to environmental and re-positive factors using the basic reproduction number $R_{0}$.

Consider the basic reproduction number $R_{0}$ as a function $R_{0}=R_{0}\left(\mu_{3},\gamma_{3}\right)$ with respect to the parameters $\mu_{3}$ and $\gamma_{3}$, and let the values of all the other parameters in the basic reproduction number $R_{0}$ be the same as those from Italy in Table 3.

Figure 5a gives the intersection line of the surface $R_{0}=R_{0}\left(\mu_{3},\gamma_{3}\right)$ and the plane $R_{0}=1$. Figure 5b shows the contour lines corresponding to the different values of $R_{0}$. The contour lines give the range of the parameters, $\mu_{3}$ and $\gamma_{3}$, which are directly related to environmental and re-positive factors. Particularly, the area on the right side of the contour line with $R_{0}=1$ can be seen as a high-risk area, which means that the disease infection will exist forever, and the area on the left side of the contour line with $R_{0}=1$ can be seen as a low-risk area, which means that the disease infection will eventually be cleared. That is, when the parameter related to environmental factors ($\gamma_{3}$) and the parameter related to re-positives ($\mu_{3}$) have their values in the low-risk area, the disease infection is controllable.

Moreover, Figure 6a,b shows that when the parameters $\gamma_{3}$ (the infection rate due to environmental vectors) or $\mu_{3}$ (the rate of re-positives) decreases, the cumulative number of infected cases also decreases significantly.

According to the above figures, the re-positive rate $\mu_{3}$ decreased 10 times, and the cumulative number of infected cases decreased 48%. This is alarming, and suggests we need to pay more attention to recovered patients and their potential infectivity. We may need to re-evaluate hospitals’ discharge criteria and current patient management systems. Therefore, we stress that caution should be exercised even after recovery from SARS-CoV-2. We believe that all discharged patients should undergo medical observation and quarantine for at least 14 days. Longer periods of observation and surveillance may be necessary. Patients who have been infected with COVID-19 virus in the past should also comply with epidemiological control measures, such as wearing a mask and keeping distance from others.

The environment-related parameter $\gamma_{3}$ decreased 10 times, and the cumulative number of infected cases decreased 58%. However, some recent research has shown that there are many ways of communication for the novel coronavirus, and communication by environmental vectors is an important link that cannot be ignored. When an infected person sneezes or coughs, droplets are released that can contaminate surfaces. If a susceptible person touches these contaminated surfaces and then touches their mouth, nose, or eyes, they can become infected. If we want to ensure that the virus does not spread on a large scale, it is very important to sanitize the environment. For normal households, sanitizing with alcohol and disinfectant water can meet the demand, and families with susceptible conditions can also purchase ozone disinfection equipment for sanitizing. For places with local epidemic break-out, professional institutions are needed for sanitization. We have obtained some crucial epidemiological parameters that need greater emphasis for the mitigation and control of the COVID-19 pandemic.

## 4. Conclusions

Inspired by the diversity and complexity of COVID-19 infections, two key factors were considered in model (1). The first is that environmental vectors polluted by virus particles may lead to disease infection, and the second is that recovered individuals who have lost sufficient immune protection may return to the exposed class (i.e., called re-positive). In the past two years especially, these factors have been increasingly observed in the spread of COVID-19 infection, and are bringing great challenges and difficulties to the control of disease infection.

Theorems 1 and 2 give a complete characterization of the global dynamics of model (1). In other words, when the basic reproduction number $R_{0}<1$, the disease-free equilibrium $Q_{0}$ is globally asymptotically stable, meaning that the disease infection will eventually be cleared. When the basic reproduction number $R_{0}>1$, model (1) is uniformly persistent, meaning that the disease infection will exist forever and become an endemic disease. Furthermore, Theorem 3 gives some sufficient conditions for the global asymptotic stability of the endemic equilibrium $Q^{*}$ of model (1). It should be mentioned here that the sufficient conditions in Theorem 3 are very conservative and can be further improved.

Model (1) was applied to the data fitting of the cumulative confirmed cases and recovered cases of COVID-19 in Japan (20 May–18 June 2022) and Italy (12 January–10 February 2022), and good predictability of the data was observed.

Furthermore, based on data from Italy (12 January–10 February 2022), it was observed that the basic reproduction number $R_{0}$ also had stronger dependence on the parameters $\mu_{3}$, $\gamma_{2}$, $\gamma_{3}$, and ρ. This indicates the necessity and practical significance of considering environmental and re-positive factors in the prevention and control of COVID-19 infection.

## Figures and Tables

**Figure 1 viruses-15-01201-f001:**
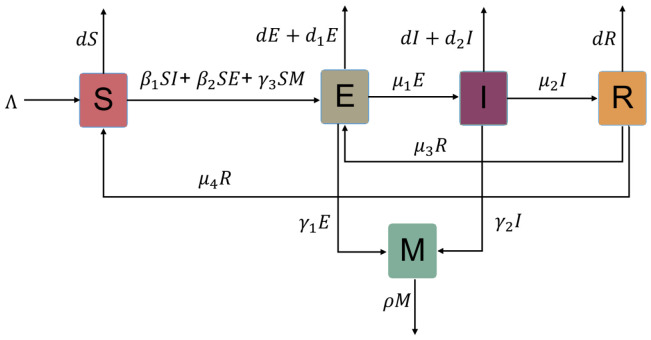
The block diagram of the interactions among *S*, *E*, *I*, *R*, and *M*.

**Figure 2 viruses-15-01201-f002:**
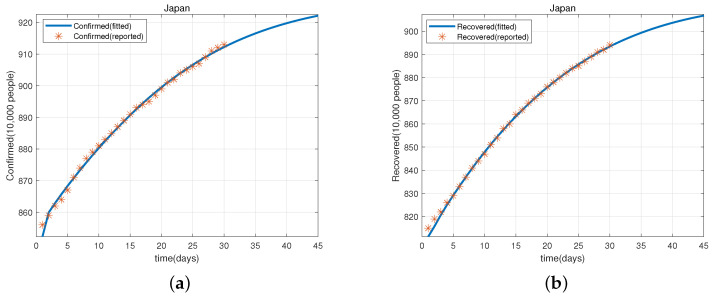
(**a**) The evolution of the cumulative confirmed cases of COVID-19 in Japan from 20 May–18 June 2022. (**b**) The evolution of the cumulative recovered cases of COVID-19 in Japan from 20 May–18 June 2022.

**Figure 3 viruses-15-01201-f003:**
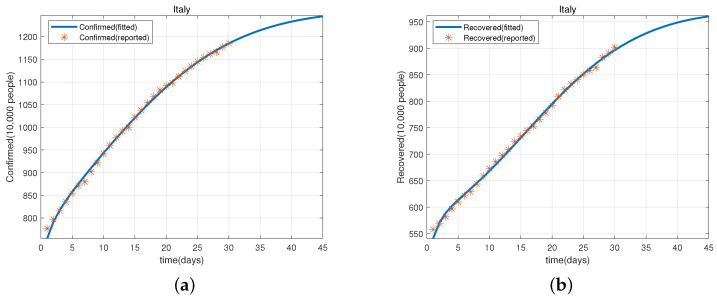
(**a**) The evolution of the cumulative confirmed cases of COVID-19 in Italy from 12 January–10 February 2022. (**b**) The evolution of the cumulative recovered cases of COVID-19 in Italy from 12 January–10 February 2022.

**Figure 4 viruses-15-01201-f004:**
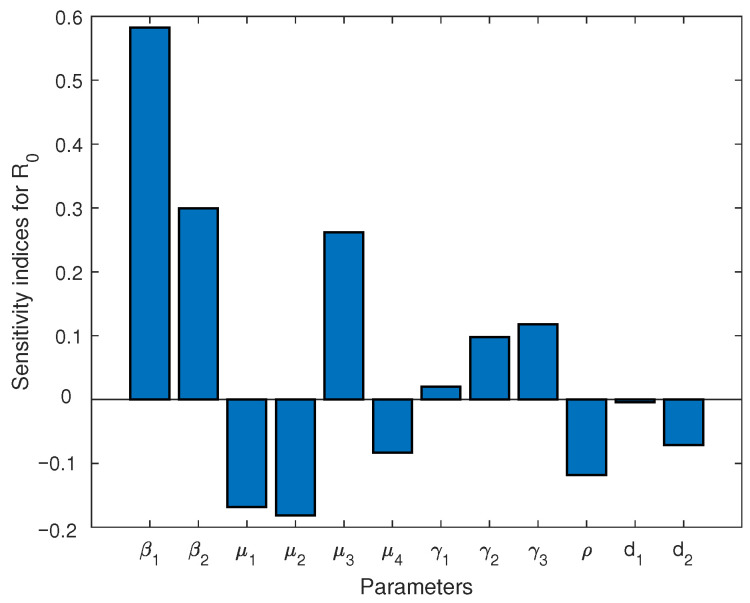
The normalized sensitivity index based on $R_{0}$ and data from Italy.

**Figure 5 viruses-15-01201-f005:**
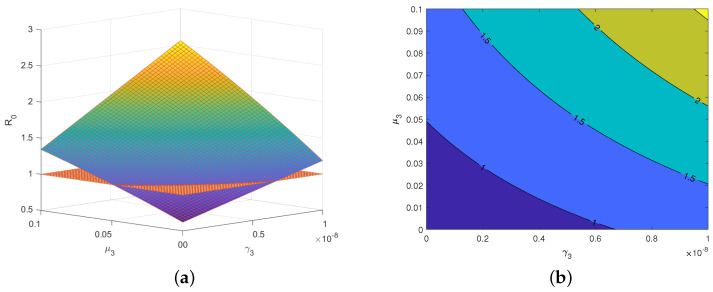
(**a**) The intersection line of $R_{0}=R_{0}\left(\mu_{3},\gamma_{3}\right)$ and $R_{0}=1$. (**b**) Contour lines corresponding to different values of $R_{0}$.

**Figure 6 viruses-15-01201-f006:**
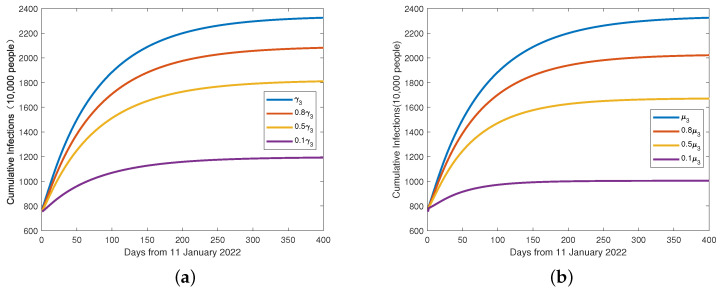
(**a**) The evolutionary relationship between the cumulative number of infected cases (*I*) and the parameter $\gamma_{3}$. (**b**) The evolutionary relationship between the cumulative number of infected cases (*I*) and the parameter $\mu_{3}$.

**Table 1 viruses-15-01201-t001:** Biological meanings of the parameters in model (1.1).

Parameter	Description	Unit
$\beta_{1}$	Infection rate of infected individuals	/10,000 persons × day
$\beta_{2}$	Infection rate of exposed individuals	/10,000 persons × day
$\gamma_{1}$	Rate of virus release into the environment by exposed individuals	10,000 viruses/10,000 persons × day
$\gamma_{2}$	Rate of virus release into the environment by infected individuals	10,000 viruses/10,000 persons × day
$\gamma_{3}$	Infection rate due to environmental vectors	/10,000 viruses × day
ρ	Rate of decay or clearance of free virus particles in environmental vectors	/day
Λ	Population input rate	10,000 persons/day
$\mu_{1}$	Rate at which exposed individuals become infected	/day
$\mu_{2}$	Recovery rate of infected people	/day
$\mu_{3}$	Rate of re-positives	/day
$\mu_{4}$	Rate at which recovered individuals without immune protection return to the susceptible class	/day
*d*	Natural death rate	/day
$d_{1}$	Death rate of exposed individuals caused by virus	/day
$d_{2}$	Death rate of infected individuals caused by virus	/day

**Table 2 viruses-15-01201-t002:** The estimated values of parameters in the model.

Para-Meter	Definitions	Value (Japan)	Value (Italy)	Unit	Source
$\beta_{1}$	See Table 1	$2.744\times 10^{-7}$	$3.195\times 10^{-6}$	/10,000 persons × day	LSM
$\beta_{2}$	See Table 1	$6.766\times 10^{-7}$	$5.123\times 10^{-6}$	/10,000 persons × day	LSM
$\gamma_{1}$	See Table 1	0.6513	0.3966	10,000 viruses/10,000 persons × day	LSM
$\gamma_{2}$	See Table 1	0.2791	0.6091	10,000 viruses/10,000 persons × day	LSM
$\gamma_{3}$	See Table 1	$1.559\times 10^{-8}$	$5.235\times 10^{-7}$	/10,000 viruses × day	LSM
ρ	See Table 1	0.8901	0.5957	/day	LSM
Λ	See Table 1	0.3916	0.3358	10,000 persons/day	LSM
$\frac{1}{\mu_{1}}$	Incubation period	5.2	5.2	day	[3]
$\frac{1}{\mu_{2}}$	Infectious period	20	20	day	[60]
$\mu_{3}$	See Table 1	0.03051	0.09186	/day	LSM
$\frac{1}{\mu_{4}}$	Time of immune protection	180	180	day	[61]
$\frac{1}{d}$	Mean lifetime	84×365	83×365	day	WHO
$d_{1}$	See Table 1	$2.025\times 10^{-4}$	$3.371\times 10^{-4}$	/day	LSM
$d_{2}$	See Table 1	$1.425\times 10^{-3}$	$2.512\times 10^{-3}$	/day	[62]

**Table 3 viruses-15-01201-t003:** Predicted values and reported data for confirmed cases in Japan.

Data	Cumulative Confirmed Cases (Japan)	
	Reported	Predicted
6.19	917.2745	912.8151
6.20	920.0541	913.6510
6.21	923.5918	914.4373
6.22	927.3191	915.1782
6.23	930.9861	915.8729
6.24	931.5674	916.5263
6.25	933.2259	917.1391
6.26	934.6495	917.7133
6.27	935.6060	918.2511
6.28	936.5433	918.7538

**Table 4 viruses-15-01201-t004:** Predicted values and reported data for recovered cases in Japan.

Data	Cumulative Recovered Cases (Japan)	
	Reported	Predicted
6.19	897.7861	894.7521
6.20	900.3232	895.9912
6.21	903.6874	897.1613
6.22	905.2890	898.2644
6.23	908.7801	899.3047
6.24	910.2259	900.2809
6.25	914.0274	901.1989
6.26	916.9366	902.0616
6.27	917.9099	902.8693
6.28	918.6873	903.0266

**Table 5 viruses-15-01201-t005:** Predicted values and reported data for confirmed cases in Italy.

Data	Cumulative Confirmed Cases (Italy)	
	Reported	Predicted
2.11	1201.3631	1191.4833
2.12	1212.1109	1198.4161
2.13	1220.3330	1203.0440
2.14	1228.5675	1209.3611
2.15	1236.4451	1214.3588
2.16	1242.5474	1218.0709
2.17	1250.5343	1223.4792
2.18	1255.3398	1226.6014
2.19	1260.7098	1230.4304
2.20	1266.1773	1233.9731

**Table 6 viruses-15-01201-t006:** Predicted values and reported data for recovered cases in Italy.

Data	Cumulative Recovered Cases (Italy)	
	Reported	Predicted
2.11	908.0136	904.2121
2.12	918.9429	911.0671
2.13	927.6892	917.8210
2.14	934.5987	923.4734
2.15	940.2540	928.2089
2.16	946.3380	933.3926
2.17	953.3268	938.6280
2.18	960.2908	942.6809
2.19	966.0380	946.5611
2.20	970.5147	949.2560

**Table 7 viruses-15-01201-t007:** MAPE/RMSPE evaluation standard.

MAPE/RMSPE	Predictive Ability
<10%	Precision prediction
10–20%	Good prediction
20–50%	Reasonable prediction
>50%	Unreasonable prediction

**Table 8 viruses-15-01201-t008:** MAPE/RMSPE values and their predictive ability.

Data Type	MAPE	Predictive Ability	RMSPE	Predictive Ability
Confirmed cases in Japan	1.40%	Precision prediction	1.56%	Precision prediction
Reovered cases in Japan	1.08%	Precision prediction	1.24%	Precision prediction
Confirmed cases in Italy	1.81%	Precision prediction	1.99%	Precision prediction
Recovered cases in Italy	1.38%	Precision prediction	1.55%	Precision prediction

**Table 9 viruses-15-01201-t009:** The normalized sensitivity index based on $R_{0}$ and data from Italy.

Parameter	$\beta_{1}$	$\beta_{2}$	$\mu_{1}$	$\mu_{2}$	$\mu_{3}$	$\mu_{4}$
Sensitivity Index	0.5827	0.2993	−0.2501	−0.1816	0.2618	−0.0833
Parameter	$\gamma_{1}$	$\gamma_{2}$	$\gamma_{3}$	ρ	$d_{1}$	$d_{2}$
Sensitivity Index	0.0200	0.0979	0.1180	−0.1180	−0.0041	−0.0713

## Data Availability

The data used for the numerical simulation analysis are from the Johns Hopkins University Center for Systems Science and Engineering, https://github.com/CSSEGISandData/COVID-19 (accessed on 1 July 2022).

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
