# Peer review of "A COVID-19 Infection Model Considering the Factors of Environmental Vectors and Re-Positives and Its Application to Data Fitting in Japan and Italy"

_viruses, 2023, doi:10.3390/v15051201_

Round 1

Reviewer 1 Report

The authors considered the transmission dynamics of COVID-19's infection based on differential equations and numerical analysis. In model (1), the two important factors are taken into account, one is the accumulation of SARS-CoV-2 in environmental vectors which has possibility to result in new infections, and another one is "Fuyang" (i.e. the recovered individuals who have lost sufficient immune protection may still return to the exposed class). These two factors have been widely observed in the infection of COVID-19 and  brought great difficulties to the control of COVID-19's infection. Model (1) has reliable biological expressions. The computing of the basic reproduction number and the proofs of the theoretical results for global dynamics (i.e., global stability and uniform persistence) of model (1) are rigorous and seems true, which give possible prediction and estimation for the evolution of the infection of COVID-19. Model (1) is used to the case studies by the data of Italy and Japan, which shows good prediction effects in short time range. Further, based on sensitivity analysis, it is visually shown  that the accumulation of live virus particles in the environment and "Fuyang" are also important factors to the control of the infection. The modelling method and the results are interesting, for which reasons it will be a valuable addition to the literature.

I recommend the publication of the paper, but the following modifications are needed. 

The work Fuyang in all the paper should be revised as Re-positive.

In the abstract, healthy individuals to susceptible individuals.

In all the paper, $R_0$ to $\mathscr{R}_0$.

In Tables 1 and 2, /10000 people/day'' to /10000 people$\times$day.

Please remove the word on in the sentence "considering on the diversity of COVID-19 transmission routes [41]".

Page 4 Line 14: any equilibrium to an equilibrium''.

Page 4 Line 4 from bellow: the disease-free equilibrium of model'' to the disease-free equilibrium $Q_0$ of the model''.

In page 5, correct the work Fuether in the sentence “Let us fuether show that …”

Page 7: It seems that there are some miscalculations in $\sigma (Q_ {22} $)~and~$ \sigma (Q_ {33})$ . Please check them carefully.

Page 11: In the definitions of "MAPE" and "RMSPE", $\times100 $'' to $\times100\%$.

Pages 15 and 17: The formats of the references are not consistent with he requirements of the journal, and I strongly suggest that the authors make corrections carefully.

Author Response

Please see attached PDF file.

Reviewer 2 Report

This article deals with an ODE model for Covid-19, integrating several processes such that reinfection and something called 'Fuyang'. The author describe and analyse a mathematical model, then apply it to Italy and Japan.

I'm a bit confused with this article, because I think there is some potential that should be more deeply exploited. My main concern is that it is not clear what new knowledge is brought to the public. To illustrate my concern, the introduction only states two goals (p3), which are mostly technical: the analysis of the mathematical model, and the application to two countries. It would have been more interesting to develop on the vector and 'Fuyang' contamination, as those processes are still unclear in 2023. Furthermore, the conclusion about the sensitivity are interesting, but should be discussed more, because it is unclear what could (and what could not) be concluded.

Regarding to the mathematical content, the analysis is well conducted, but I have a concern about what should the model really be. I have also a concern about some restrictive hypothesis: Theorem 2.3 apply under two conditions H1 and H2. How often are those hypothesis verified ? Do they apply for Italy and Japan ?

About the model itself: p2, you write ''exposed individuals [...] are still contagious". But they just got infected, how can they 'still' be contagious ? It is unclear if class E is an exposed class as in other SEIR models (which would be supported by table 1 and parameter mu_1) or something different. If this is what was meant, then the model should not have a b_2SE term in contamination. If this is different, this should be clarified, and not using an incubation class, which will imply some delay in the dynamics in the use cases, should be justified.

Regarding to the application: the effects of mu_3 or gamma_2, the confidence in the model should be discussed with more details. It is important to understand how much new information can be learnt (or cannot, which is a result too) from this study.

Other remarks:

There is a problem with the language, which seems to be sometimes taken from automatic translations. Most of the paperis ok, but there are a few sentences which sound very odd, and take some time to understand.There are a few 'it has' which sounds odd to me: p5 'In fact, it has from model (1) that...'. Also last sentence p 5... p12: superlative and comparative forms ('greatest', 'greater') sound misused to me.

Furthermore, you use the word 'Fuyang', which I hadn't seen before. It is used as a noun, and I can't find other documents in the internet using it the same way. I guess there is a translation problem here. Anyway it should be explained more in the introduction, and some references are required.

p4: I guess \Lambda is missing somewhere in the formulas for R1, R2, R3

p7: what is Pf ?

About the application, one of my concern is overfitting. There are 9 parameters that are estimated from two curves, which seems a lot. Combined with the fact that there are high sensitivity on several parameters, it is hard to be convinced that this model is accurate.

Tables 3 to 6: I don't think such details in the relative error date by date is useful. It should be synthetized.

p12: I am not sure why gamma_2 and gamma_3 have great sensitivity.

the conclusion is too basic, one would want more detail about the Fuyang and other factors. Anyway I don't think their relative part in R_0 is accurate due to the risk of overfitting.

Author Response

Please see attached PDF file.
